# Self-Organization of Polyurethane Ionomers Based on Organophosphorus-Branched Polyols

**DOI:** 10.3390/polym16131773

**Published:** 2024-06-23

**Authors:** Ilsiya M. Davletbaeva, Oleg O. Sazonov, Ilyas N. Zakirov, Alexander V. Arkhipov, Ruslan S. Davletbaev

**Affiliations:** 1Technology of Synthetic Rubber Department, Kazan National Research Technological University, 68 Karl Marx str., Kazan 420015, Russia; sazonov.oleg2010@gmail.com (O.O.S.); zakirovilyas1996@gmail.com (I.N.Z.); 2Institute of Electronics and Telecommunications, Peter the Great St. Petersburg Polytechnic University, 29 Polytechnicheskaya str., St. Petersburg 195251, Russia; arkhipov@rphf.spbstu.ru; 3Material Science and Technology of Materials Department, Kazan State Power Engineering University, 51 Krasnoselskaya str., Kazan 420066, Russia; darus@rambler.ru

**Keywords:** *ortho*-phosphoric acid etherification, organophosphorus-branched polyols, polyurethane ionomers, self-organization, vapor permeability, tensile strength

## Abstract

Based on organophosphorus branched polyols (AEPAs) synthesized using triethanolamine (TEOA), *ortho*-phosphoric acid (OPA), and polyoxyethylene glycol with MW = 400 (PEG), vapor-permeable polyurethane ionomers (AEPA-PEG-PUs) were obtained. During the synthesis of AEPAs, the reaction of the OPA etherification with polyoxyethylene glycol was studied in a wide temperature range and at different molar ratios of the starting components. It turned out that OPA simultaneously undergoes a catalytically activated etherification reaction with triethanolamine and PEG. After TEOA is fully involved in the etherification reaction, excess OPA does not react with the terminal hydroxyl groups of AEPA-PEG or the remaining amount of PEG. The *ortho*-phosphoric acid remaining in an unreacted state is involved in associative interactions with the phosphate ions of the AEPA. Increasing the synthesis temperature from 40 °C to 110 °C leads to an increase in OPA conversion. However, for the AEPA-PEG-PU based on AEPA-PEG obtained at 100 °C and 110 °C, *ortho*-phosphoric acid no longer enters into associative interactions with the phosphate ions of the AEPA. Due to the hydrophilicity of polyoxyethylene glycol, the presence of phosphate ions in the polyurethane structure, and their associative binding with the unreacted *ortho*-phosphoric acid, the diffusion of water molecules in polyurethanes is enhanced, and high values of vapor permeability and tensile strength were achieved.

## 1. Introduction

Polyurethanes (PUs) represent a wide class of materials that are actively used in various industrial fields. The physical and chemical properties of polyurethanes largely depend on the polyol and isocyanate used and determine the scope of their application [1,2,3,4,5,6,7,8,9,10,11,12,13]. The properties of polyurethanes are controlled by changes in chain length, molecular weight, functionality, and the use of elements such as fluorine, phosphorus, etc. in the polyol chain [14,15,16,17,18,19,20,21,22,23,24,25,26,27]. Such a high degree of variability in the properties of polyurethane materials has ensured their popularity. The use of compounds containing ionogenic groups in the synthesis of polyurethanes makes it possible to obtain ionomers—the class of polyurethanes that have no more than 15 mol.% of ionic groups in the main chain. Ionic groups can be included in the structure of polyurethanes by using for their synthesis oligodiols or isocyanates containing ionic groups [28,29,30]. The introduction of ionic groups into a polyurethane matrix makes it possible to improve the dispersibility of polymers in polar solvents, increase thermal stability and mechanical strength, and endow polymers with specific properties, such as biocompatibility and diffusion properties [31,32,33,34,35,36,37,38,39,40,41,42,43,44,45,46,47,48,49,50].

Ionomers are polymeric materials that, due to the content of a small number of ionic groups, are promising for the creation of new functional materials. The ability of ionic aggregates to form quartets, sextets, and supramolecular structures of higher order leads to a high probability of their self-organization into ion clusters and ion-conducting channels [51].

There are some morphological models in the literature to explain the microstructure of ionomers. The most accepted of them is the model of multiplets [52] and clusters [53], according to which ionic compounds included in the polymer structure establish interactions in discrete regions. This leads to the formation of ionic aggregates (multiplets), which interact with each other, forming clusters. There is also a model of transitions of ionic clusters from order to disorder [54,55], in which clusters undergo first-order transitions at a temperature below the melting temperature of the polymer, called the order–disorder temperature. A model for limiting the mobility of polymer chains adjacent to multiplets is also discussed, where chain segments located close to ionic multiplets have less mobility compared to chain segments located further from the multiplets. Each multiplet is surrounded by a region of limited mobility of the polymer chain. As a result, it becomes possible to influence the glass transition temperature, miscibility, rheological characteristics, and the mechanical, electrical, and diffusion properties of polymers. The goal of this issue is to create a community of authors and readers to discuss the latest research and develop new ideas and technologies in this area of research.

In [56,57,58,59], polyurethane ionomers were obtained using aminoethers of *ortho*-phosphoric acid (AEPA) as a polyol component. It was found that AEPA-PPG, synthesized using triethanolamine (TEOA), *ortho*-phosphoric acid (OPA), and polyoxypropylene glycol (PPG) with MM = 1000, is a branched structure that contains phosphate ions (Figure 1). The AEPA-PPG-PU with an ionomeric nature was investigated as a material for vapor-permeable and pervaporation membranes.

Polyurethanes based on AEPAs were also synthesized, where hydrophobic PPG was replaced by hydrophilic polyoxyethylene glycol with MW = 400 (PEG). It turned out that polyurethanes obtained using AEPA-PEG (AEPA-PEG-PU) exhibit significantly higher pervaporation characteristics when separating aqueous alcohol solutions in comparison with AEPA-PPG-PU. However, the patterns of formation of the chemical and supramolecular structure of AEPA-PEG, the influence of temperature conditions for the synthesis of AEPA-PEG on the properties of AEPA-PEG-PU, and their relationship with the diffusion properties of the resulting polyurethanes have not been studied.

## 2. Materials and Methods

### 2.1. Solvents and Reagents

Information about the solvents and reagents used can be found in [59].

### 2.2. Synthetic Procedures

#### 2.2.1. General Procedure for Synthesis of Aminoethers of *Ortho*-Phosphoric Acid (AEPA-PEG)

For the synthesis of aminoethers of *ortho*-phosphoric acid (AEPA-(2–9)-PEG), the following components were used: triethanolamine, *ortho*-phosphoric acid, and polyoxyethylene glycol with MM = 400 at molar ratios [TEOA]:[OPA]:[PEG] = 1:(2–9):(6–20). The synthesis was carried out in one stage. The calculated amount of the *ortho*-phosphoric acid and PEG was placed in a round bottom flask. For two hours, the water initially contained in an 85% solution of *ortho*-phosphoric acid was distilled off with constant stirring at a temperature of 80 °C and vacuum at a residual pressure (0.2–2.0 mm Hg). Then, triethanolamine was added to this system, and, at a residual pressure (2.0 mm Hg), the catalytic reaction of OPA etherification was carried out at a temperature of 80 °C for 1 h. Upon completion of the synthesis, AEPA-PEG was poured into a container with a ground-in lid.

#### 2.2.2. General Procedure for Synthesis of Polyurethanes Based on Aminoethers of *Ortho*-Phosphoric (AEPA-(2–9)-PEG-PU)

The calculated amount of synthesized AEPA-(2–9)-PEG was mixed with polyisocyanate (PIC) in a ratio of 1 part by weight/0.98 parts by weight, which corresponds to 2 g of AEPA-(2–9)-PEG and 1.8 g PIC. Toluene was added to the resulting system to obtain a solution containing 80% non-volatile components. The reaction was carried out in solution, and the resulting system was poured into a petri dish. The resulting polyurethane was cured at 22–25 °C. For the synthesis of polyurethanes, an aromatic polyisocyanate—“Wannate PM-200”, purchased from Kumho Mitsui Chemicals, Inc. (Shanghai, China)—was used.

### 2.3. Measurements

#### 2.3.1. Determination of the Content of *Ortho*-Phosphoric Acid by Titration

The mass fraction of phosphoric acid was determined by titration of POH groups with a standard solution of sodium hydroxide in the presence of phenolphthalein.

#### 2.3.2. NMR Spectroscopy

^1^H NMR spectra were obtained on a Bruker Avance II 500 spectrometer (Bruker, Rheinstetten, Germany) (500.13 MHz for ^1^H) using a direct BBO probehead (BB-1H-2D). The samples contained about200 μL of the investigated polymer mix and 400 μL of deuterated chloroform in standard 5 mm NMR tubes. The spectra were recorded at 30 °C. The proton chemical shift scale is referenced with respect to the residual solvent signal.

^31^P NMR spectra of the same samples were obtained on a Bruker Avance II 500 spectrometer (202.46 MHz for ^31^P) using the BBO probehead (BB-1H-2D).

#### 2.3.3. Light-Scattering of Solution

Particle size determination experiments were carried out on a Malvern Zetasizer Nano ZS (Malvern, UK). The device is equipped with the 4 mV helium–neon laser, which operates at a wavelength of 632.8 nm. The light scattering angle is 173°. Experiments were carried out at 25 °C in disposable plastic cuvettes with an optical path length of 1 cm. Deionized water and acetone were used as solvents. The instrument error is 5%, reduced to 1% by increasing the duration of the experiment to 600 s. Before measurements, the test samples were additionally filtered with a Millipore filter (Merck, Darmstadt, Germany) to remove dust.

Fourier transform infrared spectroscopy analysis (FTIR), water vapor permeability (WVP) measurements, water adsorption, measurements of the surface tension, thermomechanical analysis (TMA), mechanical loss tangent measurements (MLT), thermal gravimetric analysis (TGA), and tensile stress—strain measurements were all carried out.

Information about these methods used can be found in [58,59].

## 3. Results

### 3.1. Study of the OPA Etherification

In [58], the role of associative interactions of tertiary amines and PPG in the catalytically activated low-temperature etherification of OPA was discovered and proven. According to our research [59], the use of PEG instead of PPG also makes it possible to synthesize the corresponding aminoethers of *ortho*-phosphoric acid (AEPA-PEG). This circumstance requires an explanation of the chemical processes that occur during the synthesis of AEPA-PEG.

According to the analysis of the FTIR spectrum (Figure 2), in the TEOA-OPA-PEG system, *ortho*-phosphoric acid enters into an etherification reaction. Thus, high-intensity bands at 880 and 960 cm^−1^, corresponding to phosphate ions in the OPA composition, disappear, and bands appear at 950 and 1010 cm^−1^. The fact that OPA is practically absent in the composition of AEPA-(3–5)-PEG can be judged from the FTIR spectra.

The characteristic band for OPA is 950 cm^−1^ (the P-O bond in the P-OH group). For AEPA-(3–5)-PEG this band is practically absent. For AEPA-6-PEG, the band at 950 cm^−1^ appears, but with a slight shift to the right. The appearance of the band at 950 cm^−1^ may be due to the fact that with a sixfold excess of OPA relative to TEOA during the synthesis of AEPA-6-PEG, part of the OPA remains in the unreacted initial state. However, a slight shift to the right is a consequence of the associative interaction of free OPA with the phosphate ions of AEPK-6-PEG.

Measurements of OPA conversion during the synthesis of AEPA-PEG were carried out. The variable values were the molar excess of OPA and PEG relative to TEOA in the ratio [TEOA]:[OPA]:[PEG]. The synthesis temperature also changed.

According to the data shown in Figure 3a, with a threefold molar excess of OPA relative to TEOA (curve 1, molar ratio [TEOA]:[OPA]:[PEG] = 1:3:3), the conversion of OPA is lower compared to the reaction system based on [TEOA]:[OPA]:[PEG] = 1:3:6. That is, an increase in the mole fraction of PEG leads to an increase in OPA conversion. For AEPA-5-PEG (curve 3), the conversion of OPA was lower compared to AEPA-3-PEG (curve 2).

If we consider a series of syntheses of AEPA-PEG with a fivefold molar excess of PEG relative to TEOA (Figure 3b), we can notice that by increasing the molar excess of PEG in the system [TEOA]:[OPA]:[PEG] = 1:5:(8–20) from 8 to 20, the OPA conversion increases from 23 to 46%, that is, almost 2 times.

To confirm the involvement of the tertiary amine in the catalytic etherification of *otho*-phosphoric acid with polyoxyethylene glycol, triethanolamine was replaced by triethylamine (TEA). The reaction was carried out at a temperature of 60 °C, which is lower than the boiling point of TEA under used reaction conditions at a residual pressure of 2.0 mm Hg. Art. According to Figure 3c, the etherification reaction in this case proceeds at a high speed, and the OPA conversion reaches 49%. When the temperature rises to 60 °C, the TEA flies away, and the etherification reaction of OPA with polyoxyethylene glycol no longer occurs. The experimental data obtained confirm the need to use a tertiary amine to carry out the etherification of OPA with polyoxyethylene glycol. The same Figure 3c (curve 4) shows the time dependences of OPA conversion, where PPG was additionally introduced into the reaction system based on [TEOA]:[OPA]:[PEG] = 1:5:6 in an amount of 0.1 mol relative to 1 mole TEOA. The use of PPG in combination with PEG led to a decrease in OPA conversion. That is, PPG in the presence of PEG does not take part in the catalytic process.

In the case of the TEOA-OPA-PEG system, the reaction occurs at a higher rate but with a lower OPA conversion. Thus, at molar ratios [TEOA]:[OPA]:[PEG] = 1:3:3, 1:3:6, and 1:5:6, the conversion of OPA already practically reaches a plateau in the first minute of the reaction and ranges from 27 to 33.5%.

With an increase in the reaction system [TEOA]:[OPA]:[PEG] = 1:5:(8–20) molar excess of PEG relative to TEOA ([PEG]:[TEOA]) from 8 to 20, the nature of the dependence of OPA conversion on molar ratio [PEG]:[TEOA]. With an increase in the [PEG]:[TEOA] ratio from 6 to 8, the conversion values decrease from 36% to 22%, but with a subsequent increase in [PEG]:[TEOA] from 8 to 20. The OPA conversion increases to 46%.

The results of kinetic studies allow us to conclude that PEG, similar to PPG, takes part in the OPA etherification reaction catalytically activated by a tertiary amine. For the reaction system TEOA-OPA-PEG, interaction is possible according to the scheme shown in Figure 4, according to which OPA simultaneously participates in the etherification reaction with TEOA and PEG. After TEOA is fully involved in the etherification reaction, excess OPA does not react with the terminal hydroxyl groups of AEPA-PEG or the remaining amount of PEG. In our previous studies [58], it was experimentally shown that the rate constant of the OPA etherification reaction with triethanolamine exceeds the rate constant of the OPA etherification reaction with polyoxypropylene glycol.

When constructing the chemical structure of AEPA-PEG, shown in Figure 4, the possibility of proton abstraction from the phosphate group remaining in the unreacted state was taken into account. Proton abstraction can occur due to the ability of the PEG macrochain to fold into the crown conformation and capture a proton into the formed cavity [60].

In addition, thanks to the kinetic studies carried out, it can be concluded that for all AEPA-(3–9)-PEG, regardless of the [OPA]:[TEOA] molar ratio used in their synthesis, the structure presented in Figure 4 will be predominant.

Measurements of the particle size of AEPA-(3–5)-PEG were carried out in water (Figure 5a) and in acetone solutions (Figure 5b).

The particle sizes measured in water (Figure 5a) turned out to be the maximum for AEPA-4-PEG. The most likely reason for their relatively large sizes is the involvement of unreacted OPA in the associated interaction with AEPA-4-PEG. As a result, *ortho*-phosphoric acid remains in its structure and does not diffuse into the water. The size of AEPA-5-PEG, obtained with a higher content of OPA in comparison with AEPA-4-PEG, decreases due to the fact that part of the unreacted OPA ceases to be retained by associative interactions in the structure of AEPA-5-PEG and passes into the surrounding aqueous environment. Thus, the amount of *ortho*-phosphoric acid associated in the structure of AEPA-5-PEG does not exceed two molecules of OPA.

In acetone solutions, the particle sizes of AEPA-(3–5)-PEG and the width of the size distribution curves also depend on the [OPA]:[TEOA] molar ratio. However, in contrast to an aqueous solution, the largest sizes in an acetone medium are observed for AEPA-3-PEG.

Further studies were carried out using ^1^H NMR and ^31^P NMR spectroscopy (Figure 6). The ^1^H NMR spectra for OPA show one narrow resonance signal on protons with a chemical shift δ = 4.7 ppm. (Figure 6a). For AEPA-(3–5)-PEG, the signal of the protons of the *ortho*-phosphoric acid groups shifts from δ = 7 ppm for AEPA-3-PEG and up to δ = 8 ppm for AEPA-5-PEG. For AEPA-4-PEG and AEPA-5-PEG, this signal expands due to its splitting. With an increase in the mole fraction of OPA during the synthesis of AEPA-(3–5)-PEG, a change in shape and a shift to a weaker field (by 0.5 ppm) of low-intensity resonance signals with a chemical shift δ = 3.7 ppm, corresponding to the proton, are also observed as part of the hydroxyl groups of PEG. According to the ^1^H NMR spectra analysis, phosphate ions in AEPA-(3–5)-PEG involve unreacted OPA molecules in associative interactions. The shift of the proton signal to the region of a weak magnetic field is most likely a consequence of their capture by the cavity formed by the open-chain analogue of crown ether, which is PEG, in accordance with the diagram shown in Figure 4.

In the ^31^P NMR spectra (Figure 6b) of AEPA-(3–4)-PEG, intense resonance signals of ^31^P nuclei are observed at δ = −0.4 ppm in the OPA, and weak signals corresponding to ^31^P in the phosphate ions are observed at δ = −0.5 ppm. The results of kinetic studies, according to which a relatively low conversion of phosphate groups is observed during the synthesis of AEPA-(3–4)-PEG, is also confirmed by the low intensity of signals in the region δ = 0.7 ppm, corresponding to the P-O-C bond. As the OPA content increases to AEPA-5-PEG, the signal corresponding to the P-O-C bond shifts to δ = 1.0 ppm. There is also a shift of the signals corresponding to ^31^P in the phosphate ions and in the OPA to a stronger field. This shift of signals and their splitting in the region δ = 0.7 ppm reflects the involvement of the OPA not only in chemical but also in specific associative interactions.

The results obtained correlate with the data of thermogravimetric analysis (Figure 7). Thus, the onset of the thermal decomposition of polyoxyethylene glycol occurs at around 275 °C. For a solution of OPA in PEG, obtained at the molar ratio [OPA]:[PEG] = 1:3, the onset of weight loss is observed at 158 °C, corresponding to the boiling point of OPA, and the beginning of intense mass loss corresponds to the temperature of thermal decomposition of OPA, located in the region 213 °C. Since the observed boiling and decomposition temperatures of OPA in the PEG medium correspond to the thermal characteristics of pure OPA, we can conclude that OPA exhibits relatively weak intermolecular interactions with PEG molecules. For AEPA-3-PEG (Figure 7, curve 3), the temperature of the onset of weight loss is slightly lower in comparison with PEG but significantly higher in comparison with the OPA solution in PEG. That is, due to the tendency of phosphate groups to strong associative interactions and the structural features of AEPA-PEG, the unreacted part of OPA is so firmly integrated into the structure of AEPA-3-PEG that the residual content of OPA does not evaporate and does not decompose when the boiling and decomposition temperatures of OPA are reached. For AEPA-6-PEG, the thermal stability begins to decrease due to the weak binding of part of the unreacted OPA to the structure of the amino ether of *ortho*-phosphoric acid.

For the synthesis of polyurethanes, AEPA-(2–9)-PEG and the aromatic polyisocyanate (PIC) “Wannate PM-200” were used. The reaction of urethane formation is widely known [1,2,3,4,5,6,7,8,9,10,11,12,13], and in this work, it involves the interaction of the terminal hydroxyl groups of AEPA-(2–9)-PEG and the isocyanate groups of PIC.

According to Figure 8, the increase in the strength of the polyurethane samples as the [H_3_PO_4_]:[TEOA] ratio increases from 3 to 5 may be due to intermolecular interactions between the ionogenic groups of AEPA-PEG. A further increase in the OPA content in the AEPA-PEG leads to the appearance of OPA not involved in the associative interactions. As a result, a significant decrease in the strength of polyurethanes due to screening interchain interactions and plasticizing the material is observed.

It should be noted that the strength of AEPA-(3–5)-PEG is high and reaches values of 47–53 MPa. For comparison, the strength of polyurethane ionomers studied in [61] is in the range of 20–30 MPa.

### 3.2. Effect of the AEPA-5-PEG Synthesis Temperature on the Some AEPA-5-PEG-PU Properties

Figure 9 shows the time dependences of OPA conversion for the reaction system [TEOA]:[OPA]:[PEG] = 1:5:6 (AEPA-5-PEG), measured under different temperature conditions. It turned out that changing the temperature conditions from T_synthesis_ = 40 °C to T_synthesis_ = 100 °C leads to an increase in conversion from 29% to 40%. With a further increase in the synthesis temperature to T_synthesis_ = 110 °C, the conversion of OPA already exceeds 50%.

In order to assess the effect of the synthesis temperature conditions on the structure of AEPA-5-PEG and the AEPA-5-PEG-PU obtained using them, studies were carried out on the surface-active properties of AEPA-5-PEG (Figure 10), synthesized at different temperatures, and the vapor permeability of the corresponding AEPA-5-PEG-PU (Figure 11).

Measurements of surface tension isotherms (Figure 10) made it possible to establish that with an increase in the synthesis temperature from 80 to 110 °C, the surface tension of AEPA-5-PEG solutions in water decreases. The obtained result confirms the conclusion that an increase in OPC conversion entails growth in the content of phosphate ions.

Accordingly, an increase in the yield of the structure presented in Figure 4 with an increase in the synthesis temperature of the AEPA-5-PEG from 40 °C to 80 °C leads to growing vapor permeability values of the polyurethanes obtained using them. With a further increase in temperature from 90 to 100 °C, the amount of unreacted OPA decreases and, in turn, leads to diminution in the vapor permeability indicators for the corresponding AEPA-5-PEG-PU (Figure 11a).

Analysis of the results obtained allows us to conclude that the associated form of OPA, built into the structure of AEPA-5-PEG-PU, has a significant effect on the diffusion of water through this polymer. Such conclusions are also confirmed by a twofold decrease in the vapor permeability coefficient from the values of WVP = 3890 g/cm^2^ for AEPA-5-PEG-PU, obtained at 80 °C and a molar ratio of [TEOA]:[OPA]:[PEG] = 1:5:6 to WVP = 1920 g/cm^2^ in 24 h, and for AEPA-5-PEG-PU, obtained at 80 °C and a molar ratio of [TEOA]:[OPA]:[PEG] = 1:5:20 (Figure 11b). As shown in Figure 3, an increase in the mole ratio of PEG relative to TEOA during the synthesis of AEPA-5-PEG leads to a noticeable increase in the conversion of OPA and, accordingly, a decrease in the content of OPA in the associated state.

According to the change in the features of the structural organization of AEPA-5-PEG obtained under different temperature conditions, the features of the supramolecular organization of AEPA-5-PEG-PU obtained using them also change (Figure 12). Thus, for AEPA-5-PEG-PU based on AEPA-5-PEG obtained at 100 °C and 110 °C, where OPA is practically absent in associated form, the temperatures of the relaxation transitions are in the region of 75 °C for AEPA-5-PEG -PU (100 °C, curve 2) and in the region of 115 °C for AEPA-5-PEG (110 °C, curve 3). Most likely, these relaxation transitions are caused by the destruction of phosphate ion clusters. In the case of AEPA-5-PEG-PU (80 °C, curve 1), the introduction of OPA in the form of associates also contributes to the formation of cluster structures. But in this case, such cluster structures have less strength and already disintegrate at 20 °C.

It should be noted that the proposed cluster structures do not have a noticeable effect on the deformation behavior of polyurethanes in the regions of high-temperature relaxations. Judging by the results of measuring vapor permeability coefficients (Figure 11), the cluster structures formed in AEPA-5-PEG-PU prevent the diffusion of water molecules. However, in the case of AEPA-5-PEG-PU (Figure 12, 80 °C, curve 1), cluster structures are already destroyed at 20 °C. For this reason, at 40 °C, due to the destruction of cluster structures and the high content of ionogenic groups, in this case, high diffusion characteristics for water molecules are observed.

### 3.3. Comparison of Surface Morphology of AEPA-6-PPG-PU and AEPA-5-PEG-PU

To establish the influence of the nature of the diol used (PEG or PPG) on the features of the supramolecular organization of the corresponding AEPA-PU, the surface morphology of the resulting polyurethanes was studied using atomic force microscopy (Figure 13). For measurements, samples of polyurethanes with an established optimal composition were used (AEPA-6-PPG-PU and AEPA-5-PEG-PU). According to the measurements carried out for AEPA-6-PPG-PU, a pronounced globular morphology is observed on the surface of the sample. The surface morphology of AEPA-5-PEG-PU differs significantly from the surface morphology of AEPA-6-PPG-PU. For AEPA-5-PEG-PU, craters appear, which are a consequence of the complex supramolecular organization of the polymer. Another feature of the manifestation of the morphology of AEPA-5-PEG-PU is the fact that the inner surface of the craters also exhibits its own morphology.

## 4. Conclusions

Based on the organophosphorus-branched polyols synthesized using polyoxyethylene glycol, vapor-permeable polyurethane ionomers were obtained. The reaction to etherification of *ortho*-phosphoric acid with polyoxyethylene glycol was studied at different molar ratios of the starting components in the presence of tertiary amines. It was found that OPA simultaneously undergoes a tertiary amine catalytically activated etherification reaction with triethanolamine and PEG. After TEOA is fully involved in the etherification reaction, excess OPA does not react with the terminal hydroxyl groups of AEPA-PEG or the remaining amount of PEG.

It was suggested that a proton can be abstracted from the phosphate group remaining in the unreacted state, resulting in the formation of phosphate ions. Proton abstraction can occur due to the ability of the PEG macrochain to fold into the crown conformation and capture a proton into the formed cavity. In the AEPA-(3–5)-PEG, unreacted orthophosphoric acid is involved in associative interactions with the phosphate ions of organophosphorus-branched polyols. Research findings indicate that the quantity of orthophosphoric acid present in the structure of AEPA-5-PEG does not exceed two molecules of orthophosphoric acid.

The hydrophilicity of PEG, the presence of phosphate ions in polyurethanes and their associative binding with unreacted *ortho*-phosphoric acid, and the diffusion of water molecules in polyurethanes are enhanced, and high values of vapor permeability and tensile strength are achieved.

## Figures and Tables

**Figure 1 polymers-16-01773-f001:**
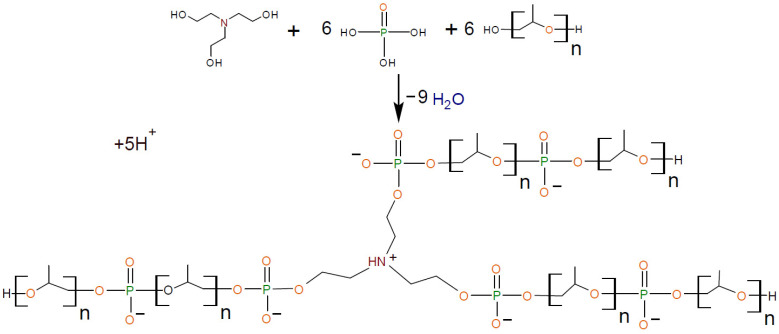
Formation of AEPA-6-PPG (n = 17).

**Figure 2 polymers-16-01773-f002:**
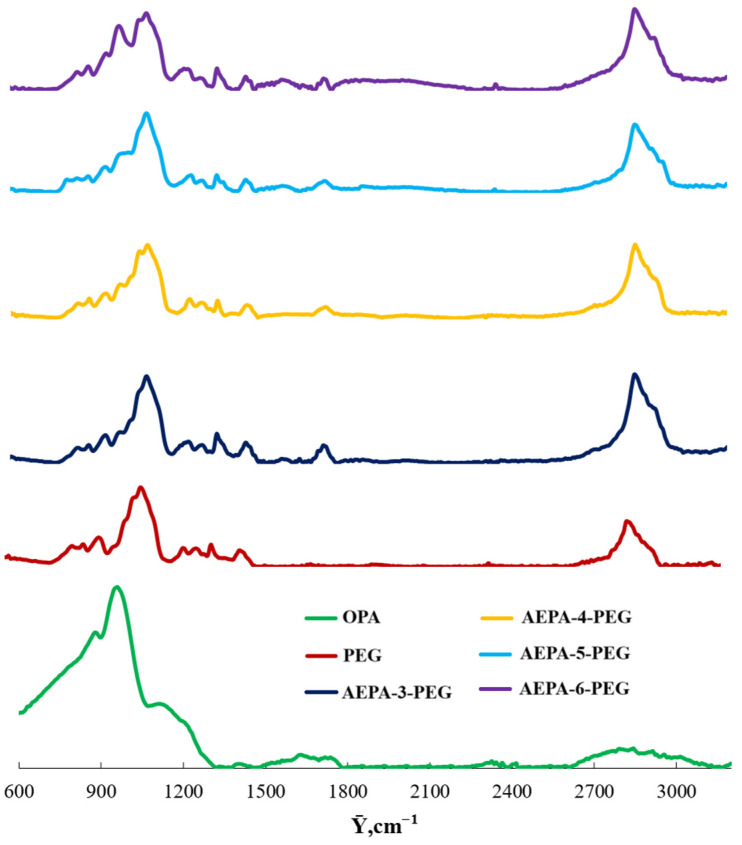
FTIR spectra.

**Figure 3 polymers-16-01773-f003:**
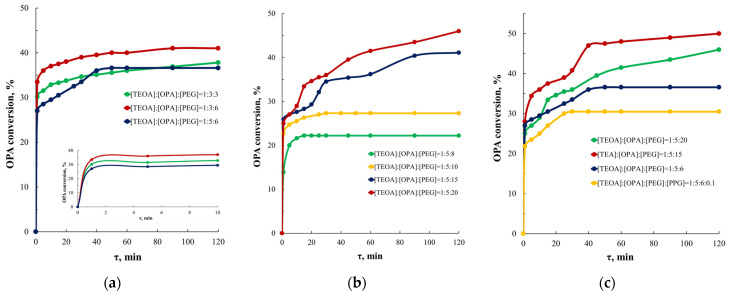
Time dependences of OPA conversion for reaction systems based on [TEOA]:[OPA]:[PEG] = 1:3÷5:3÷6 (**a**), [TEOA]:[OPA]:[PEG] = 1:5:8÷20 (**b**), [TEOA]:[OPA]:[PEG] = 1:5:6÷20 and [TEOA]:[OPA]:[PEG]:[PPG] = 1:5:6:0.1 (**c**). T_synthesis_ = 80 °C.

**Figure 4 polymers-16-01773-f004:**
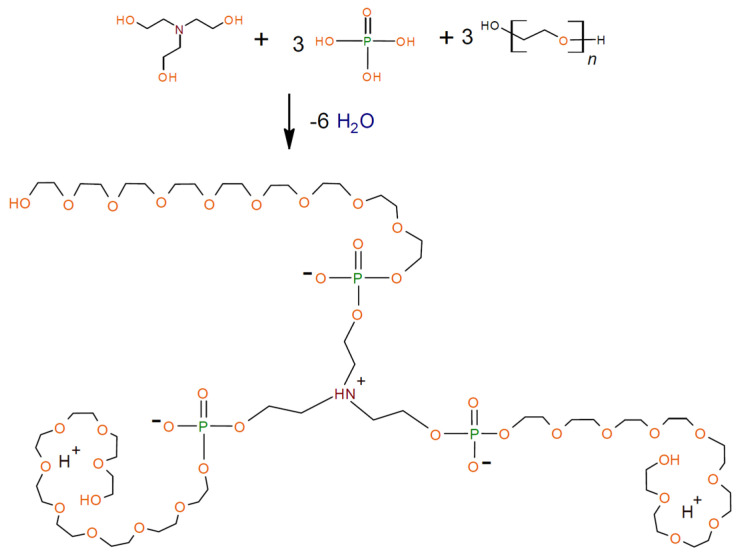
Formation of the main AEPA-(3–9)-PEG structure (n = 9).

**Figure 5 polymers-16-01773-f005:**
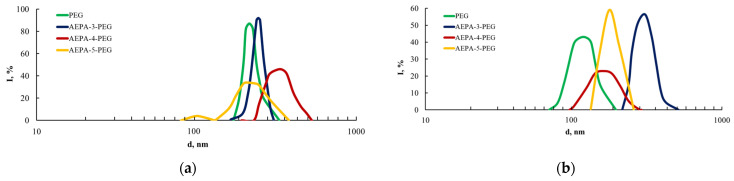
Particle size distribution in the aqueous (**a**) and acetone (**b**) solutions.

**Figure 6 polymers-16-01773-f006:**
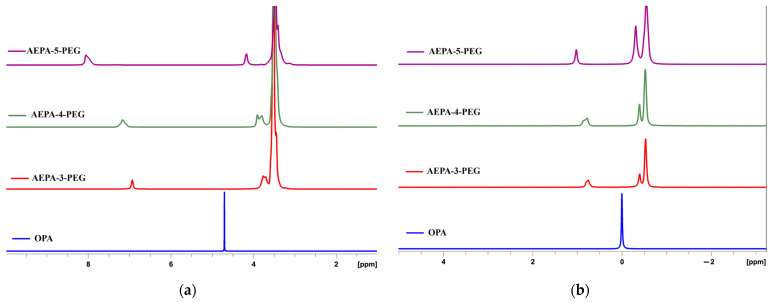
^1^H NMR (**a**) and ^31^P NMR (**b**) spectra. Solvent deuterated chloroform.

**Figure 7 polymers-16-01773-f007:**
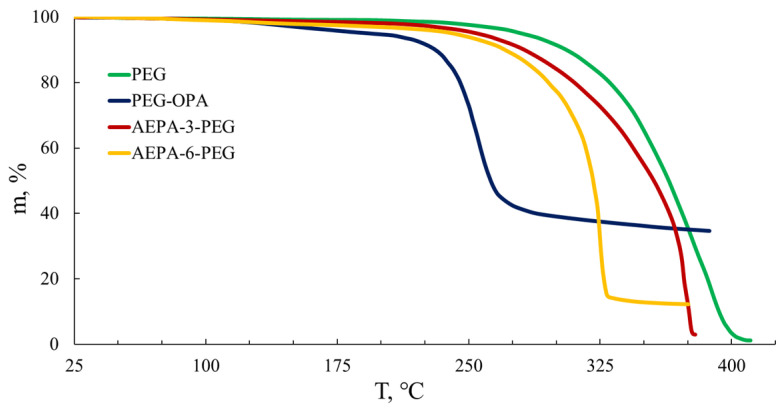
TGA curves.

**Figure 8 polymers-16-01773-f008:**
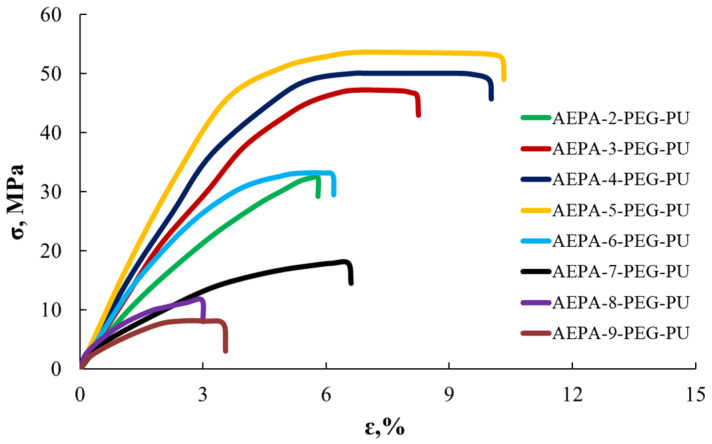
Stress—strain curves.

**Figure 9 polymers-16-01773-f009:**
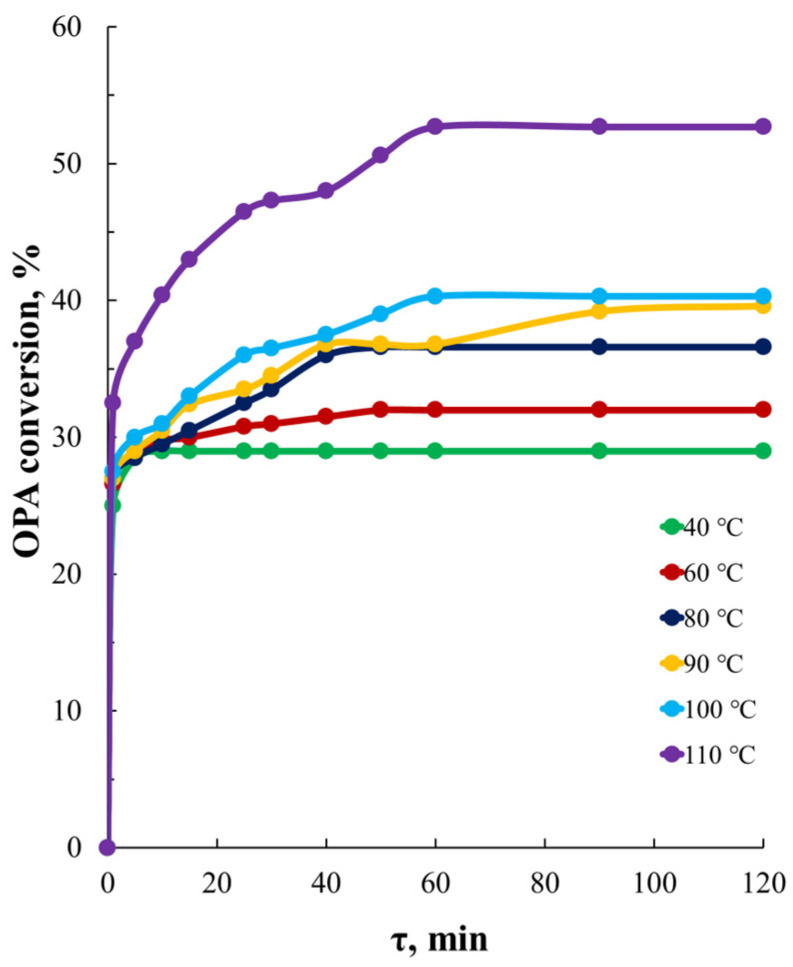
Time dependences of OPA conversion for reaction systems based on [TEOA]:[OPA]:[PEG] = 1:5:6 synthesized at different temperatures.

**Figure 10 polymers-16-01773-f010:**
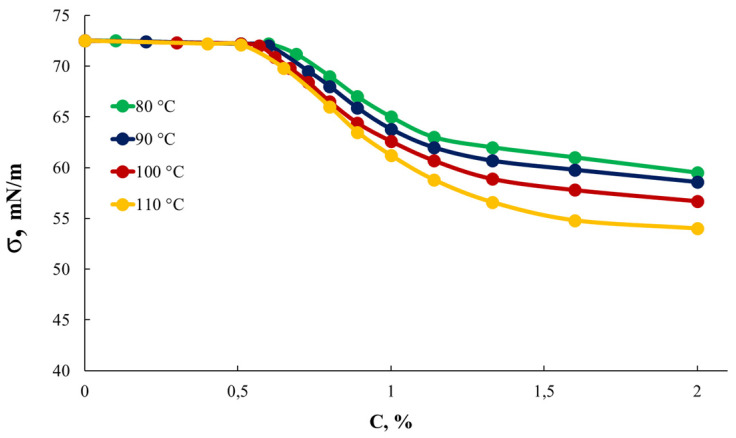
Surface tension isotherms for AEPA-5-PEG synthesized at different temperatures.

**Figure 11 polymers-16-01773-f011:**
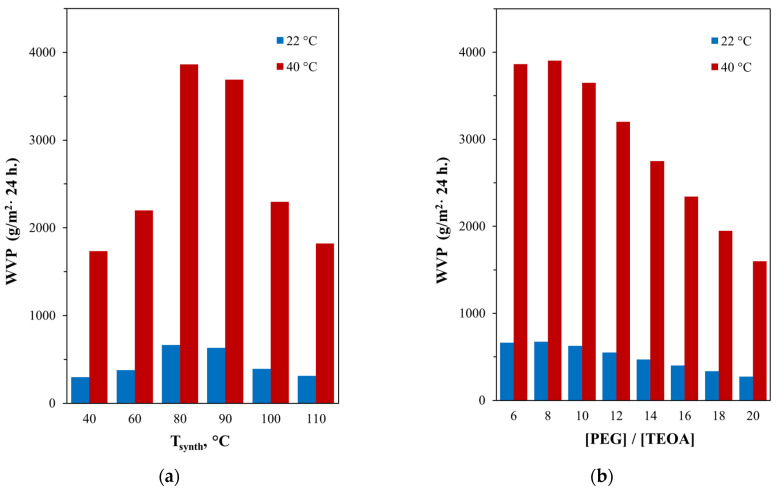
Vapor permeability coefficients for AEPA-5-PEG-PU based on AEPA-5-PEG, obtained at different synthesis temperatures (**a**) and for AEPA-5-PEG-PU depending on the molar ratio [PEG]:[TEOA], used in the synthesis of AEPA-PEG (**b**).

**Figure 12 polymers-16-01773-f012:**
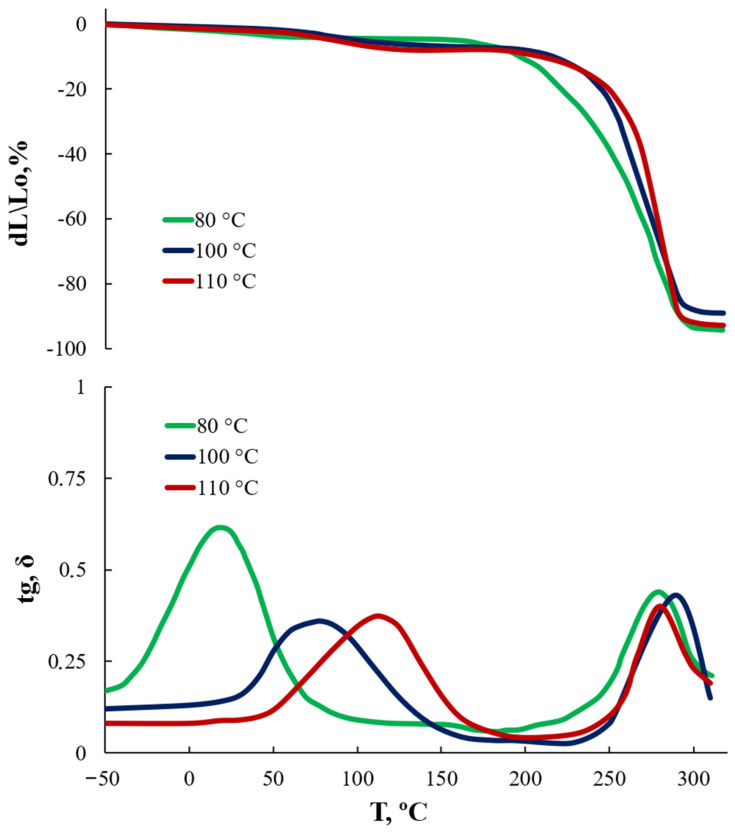
TMA and DMA curves (tg, σ) for AEPA-5-PEG-PU based on AEPA-5-PEG, synthesized at different temperatures.

**Figure 13 polymers-16-01773-f013:**
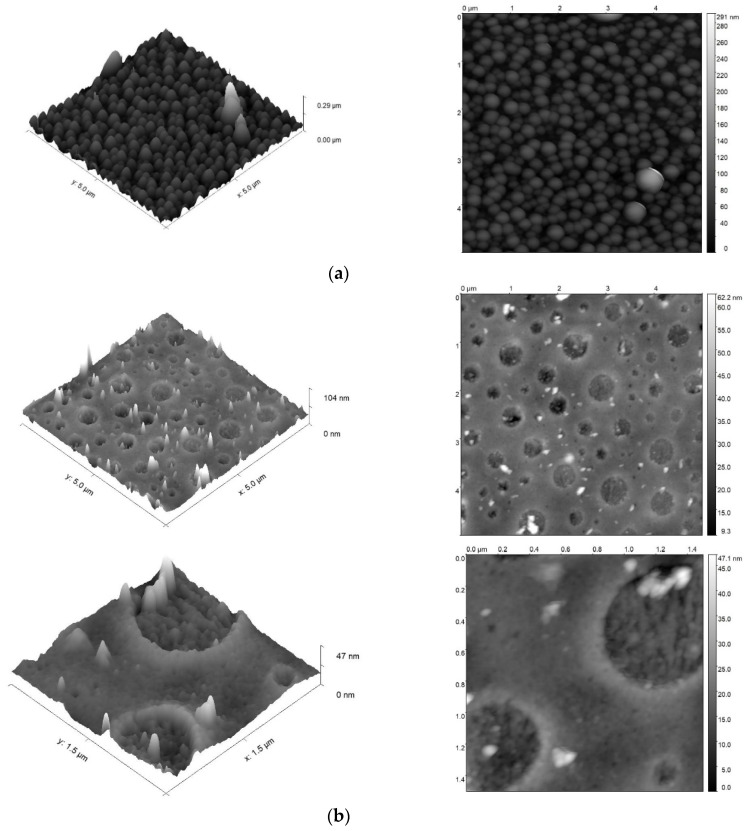
AFM images for AEPA-6-PPG-PU (**a**) and AEPA-5-PEG-PU (**b**).

## Data Availability

The original contributions presented in the study are included in the article, further inquiries can be directed to the corresponding authors.

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
