# Peer review of "Self-Organization of Polyurethane Ionomers Based on Organophosphorus-Branched Polyols"

_polymers, 2024, doi:10.3390/polym16131773_

Round 1
Reviewer 1 Report
Comments and Suggestions for Authors
The authors present a very detailed study on the preparation of triethanolamine, orthophosphoric acid and polyethylene glycol based aminoether ionomers and their use as polyols for polyurethanes for water permeable membranes. However, there is a lack of clarity with respect to multiple aspects of the reactions studied here. The following comments should help address some of these issues.
-
Authors should include any drying procedures undertaken for the polyurethane components (often the presence of moisture can cause unwanted side reactions with isocyanates). In addition, were any catalysts used for the polyurethane synthesis?
-
Please clarify what n in AEPA-n-PEG/PPG is. From figure 1, this appears to be the ratio of TEOA:OPA (1:6 in case of AEPA-6-PPG). The authors should explicitly clarify this. Further it is not clear what structures are forming here- figure 4 indicates that the phosphate ions only link the TEOA with the PEG chain yet in the case of the PPG oligomers, these phosphate ions are shown to link TEOA with PPG as well as two PPG chains. What is the reason for this supposed discrepancy and what is the correct general structure for the AEPA-n-PEG products?
-
How do the authors measure the OPA conversion for the kinetic studies? This should be noted in the materials and methods section.
-
Line 122: The authors note a shift in the FTIR band for OPA (typically seen at 950 cm-1) in case of AEPA-6-PEG. Is this due to free OPA in non-covalent interactions with the AEPA-6-PEG oligomers?
-
Line 146: “When the temperature rises to 60 °C, with TEA flies away, …” Is the temperature here incorrect? (TEA boiling point is around 90 °C and the experiment described in the immediately prior lines already uses 60 °C.
-
In all the figures showing conversion of OPA vs time, is there a reason why the y axes are inverted? It may be clearer to see the trends if the y axes increase in quantity as they go up, starting from 0 at the bottom.
-
Line 169: “After TEOA is fully involved in the etherification reaction…” By fully involved, do the authors mean fully consumed? Additionally, it may be helpful to show the kinetics of TEOA and PEG consumption along with OPA consumption for a representative case.
-
In line 183, the authors say that there is no unreacted OPA in AEPA-4-PEG yet two lines later they claim OPA is retained in the “large structure” of AEPA-4-PEG and does not “diffuse” into water. The authors should clarify what they mean by this.
-
The language in line 207 is not entirely clear.
-
If possible, the authors should prepare trisubstituted aminoether model compounds, perhaps with TEOA OPA and nonfunctional alcohols, to confirm the NMR shifts of the protons and the phosphate ion present in the aminoether linkage. This could serve to further strengthen the NMR observations.
-
For OPA consumption curves where TEOA/OPA is 1:3 (AEPA-3-PEG), OPA conversion does not exceed 40%. Shouldn’t AEPA-3-PEG then have residual OPA that evaporates/decomposes at a lower temperature in the TGA experiments? Unless there is a misunderstanding on my part, APEA-3-PEG does not correspond to a material with a starting mixture having a TEOA/OPA ratio of 1:3.
-
Line 236-237: It is not clear why associated OPA ions would increase rigidity of the PU. One may expect them to do the opposite, by screening interchain interactions and plasticizing the material.
-
Why does the strength of AEPA-n-PEG-PU start to decrease after n=5? Some hypotheses would be welcome here.
-
Authors claim that WVP of the PU membrane decreases as the synthesis temperature of the AEPA-PEG increases from 90-100 °C. However, they do not explain the increase observed from synthesis temperatures 40-80 °C.
-
In line 311-312, the authors claim that the presence of craters in the AFM profiles of AEPA-5-PEG-PU appear due to supramolecular organization of the polymer. What causes craters to appear for AEPA-5-PEG-PU but not AEPA-6-PEG-PU. Is it due to a difference in the amount of OPA.
Issues such as lack of clarity and typographical errors in some places have been mentioned in the comments above.
Author Response
Thank you for your comments and feedback, which have helped us to substantially improve our manuscript. We have carefully studied your recommendations and prepared comments on each item you mentioned in the review.
- Authors should include any drying procedures undertaken for the polyurethane components (often the presence of moisture can cause unwanted side reactions with isocyanates). In addition, were any catalysts used for the polyurethane synthesis?
Answer: Before the synthesis of AEPA-n-PEG/PPG, polyoxypropylene glycol and polyoxyethylene glycol were dehydrated using a membrane pump at 80 °C and a residual pressure of 0.7 kPa. In addition, during the synthesis of AEPA-n-PEG/PPG, the reaction mass was dehydrated under the same conditions. No additional catalysts were used for the synthesis of polyurethanes. To clarify the conditions for the synthesis of polyurethanes, additions have been made to paragraph 2.2 of the experimental part.
- Please clarify what n in AEPA-n-PEG/PPG is. From figure 1, this appears to be the ratio of TEOA:OPA (1:6 in case of AEPA-6-PPG). The authors should explicitly clarify this. Further it is not clear what structures are forming here- figure 4 indicates that the phosphate ions only link the TEOA with the PEG chain yet in the case of the PPG oligomers, these phosphate ions are shown to link TEOA with PPG as well as two PPG chains. What is the reason for this supposed discrepancy and what is the correct general structure for the AEPA-n-PEG products?
Answer: The number n in AEPA-n-PEG/PPG indicates the mole fraction of ortho-phosphoric acid relative to triethanolamine in the synthesis of the corresponding AEPA-PEG/PPG. For example, in AEPA-6-PPG, the number 6 means that the mole fraction of ortho-phosphoric acid relative to triethanolamine in the synthesis of AEPA-PEG/PPG is 6 times greater. In connection with the comment, additions have been made to the description of Figure 4: «Thus, AEPA-(3–5)-PEG is characterized by the formation in all cases of the structure of the AEPA-3-PEG shown in Figure 4.». Line 208-210.
- How do the authors measure the OPA conversion for the kinetic studies? This should be noted in the materials and methods section.
Answer: Determination of the content of ortho-phosphoric acid by titration was added to the experimental part.
- Line 122: The authors note a shift in the FTIR band for OPA (typically seen at 950 cm-1) in case of AEPA-6-PEG. Is this due to free OPA in non-covalent interactions with the AEPA-6-PEG oligomers?
Answer: In connection with the commentary, additions were made to the manuscript: «The appearance of the band at 950 cm-1 may be due to the fact that with a sixfold excess of OPA relative to TEOA during the synthesis of AEPA-6-PEG, part of the OPA remains in the unreacted initial state. However, a slight shift to the right is a consequence of the associative interaction of free OPA with the phosphate ions of AEPA-6-PEG.». Line 151-155.
- Line 146: “When the temperature rises to 60 °C, with TEA flies away, …” Is the temperature here incorrect? (TEA boiling point is around 90 °C and the experiment described in the immediately prior lines already uses 60 °C.
Answer: Indeed, TEA boiling point is around 90 °C under normal conditions. However, the reaction was carried out under reduced pressure, which was not noted. In this regard, the following additions have been made to the manuscript:«The reaction was carried out at a temperature of 60 °C, which is below the boiling point of TEA in the reaction conditions used where the residual pressure is 2.0 mm Hg». Line 172-174.
- In all the figures showing conversion of OPA vs time, is there a reason why the y axes are inverted? It may be clearer to see the trends if the y axes increase in quantity as they go up, starting from 0 at the bottom.
Answer: Corrected.
- Line 169: “After TEOA is fully involved in the etherification reaction…” By fully involved, do the authors mean fully consumed? Additionally, it may be helpful to show the kinetics of TEOA and PEG consumption along with OPA consumption for a representative case.
Answer: Indeed, a representative case should have been shown. In response to this comment, the following has been added to the text: «In our previous study [58], it was experimentally shown that the rate constant of the OPA etherification reaction with triethanolamine exceeds the rate constant of the OPA etherification reaction with polyoxypropylene glycol.». Line 200-202.
- In line 183, the authors say that there is no unreacted OPA in AEPA-4-PEG yet two lines later they claim OPA is retained in the “large structure” of AEPA-4-PEG and does not “diffuse” into water. The authors should clarify what they mean by this.
Answer: We agree with the comment. Changes have been made to the text for clarity: «The particle sizes turned out to be maximum for AEPA-4-PEG. The most likely reason for their relatively large sizes is due to the involvement of unreacted OPA in the associated interaction with AEPA-4-PEG. As a result, ortho-phosphoric acid remains in its structure and does not diffuse into the water.». Line 216-219.
- The language in line 207 is not entirely clear.
Answer: Changes have been made to the text: «According to the 1H NMR spectra analysis, phosphate ions in AEPA-(3–5)-PEG involve unreacted OPA molecules in associative interactions». Line 239-240.
- If possible, the authors should prepare trisubstituted aminoether model compounds, perhaps with TEOA OPA and nonfunctional alcohols, to confirm the NMR shifts of the protons and the phosphate ion present in the aminoether linkage. This could serve to further strengthen the NMR observations.
Answer: We ask the reviewer to give us the opportunity to begin such measurements in our further studies.
- For OPA consumption curves where TEOA/OPA is 1:3 (AEPA-3-PEG), OPA conversion does not exceed 40%. Shouldn’t AEPA-3-PEG then have residual OPA that evaporates/decomposes at a lower temperature in the TGA experiments? Unless there is a misunderstanding on my part, APEA-3-PEG does not correspond to a material with a starting mixture having a TEOA/OPA ratio of 1:3.
Answer: In accordance with the comment, we have clarified the manuscript:
- «The results obtained correlate with the data of thermogravimetric analysis (Fig. 7). Thus, the onset of thermal decomposition of polyoxyethylene glycol is around 275 °C. For a solution of OPA in PEG, obtained at the molar ratio [OPA]:[PEG] = 1:3, the onset of weight loss is observed at 158 °C, corresponding to the boiling point of OPA, and the be-ginning of intense mass loss corresponds to the temperature of thermal decomposition of OPA, located in the region 213 °C. Since the observed boiling and decomposition temper-atures of OPA in the PEG medium correspond to the thermal characteristics of pure OPA, we can conclude that OPA exhibits relatively weak intermolecular interactions with PEG molecules. For AEPA-3-PEG (Fig. 7, curve 3), the temperature of the onset of weight loss is slightly lower in comparison with PEG, but significantly higher in comparison with the OPA solution in PEG. That is, due to the tendency of phosphate groups to strong associa-tive interactions and the structural features of AEPA-PEG, the unreacted part of OPA is so firmly integrated into the structure of AEPA-3-PEG that the residual content of OPA does not evaporate and does not decompose when the boiling and decomposition temperatures of OPA are reached. For AEPA-6-PEG, the thermal stability begins to decrease due to weak binding of part of the unreacted OPA to the structure of the amino ether of or-tho-phosphoric acid.» Line 255-286.
- In addition, thanks to the kinetic studies carried out, it can be concluded that for all AEPA-(3–9)-PEG, regardless of the [OPA]:[TEOA] molar ratio used in their synthesis, the structure presented in Figure 4 will be predominant. Line 208-210.
12. Line 236-237: It is not clear why associated OPA ions would increase rigidity of the PU. One may expect them to do the opposite, by screening interchain interactions and plasticizing the material.
Answer: We agree with the comment. The following changes have been made to the manuscript: «According to Fig. 8 the increase in the strength of polyurethane samples as the [H3PO4]:[TEOA] ratio increases from 3 to 5 may be due to intermolecular interactions between the ionogenic groups of AEPA-PEG. A further increase in the OPA content in the AEPA-PEG leads to the appearance of OPA not involved in the associative interactions. As a result, a significant decrease in the strength of polyurethanes due to screening interchain interactions and plasticizing the material is observed.» Line 278-284.
- Why does the strength of AEPA-n-PEG-PU start to decrease after n=5? Some hypotheses would be welcome here.
Answer: The answer to this comment is contained in the reply to the comment 12.
- Authors claim that WVP of the PU membrane decreases as the synthesis temperature of the AEPA-PEG increases from 90-100 °C. However, they do not explain the increase observed from synthesis temperatures 40-80 °C.
Answer: Explanation added to the manuscript: «Accordingly, an increase in the yield of the structure presented in Figure 4 with an increase in the synthesis temperature of the AEPA-5-PEG from 40 °C to 80 °C leads to growing vapor permeability values of polyurethanes obtained using them. With a further increase in temperature from 90 to 100 °C, the amount of unreacted OPA decreases, and in turn, leads to diminution in vapor permeability indicators for the corresponding AEPA-5-PEG-PU (Fig. 11a).» Line 306-311.
- In line 311-312, the authors claim that the presence of craters in the AFM profiles of AEPA-5-PEG-PU appear due to supramolecular organization of the polymer. What causes craters to appear for AEPA-5-PEG-PU but not AEPA-6-PEG-PU. Is it due to a difference in the amount of OPA.
Answer: AEPA-PU samples were used for measurements, where the nature of the diols (PEG or PPG) was changed in AEPA. Figure 13(a) shows AFM images for AEPA-6-PPG-PU, the structure of which is shown in Figure 1. The AEPA-6-PEG-PU sample is not shown in Figure 13(a).
Reviewer 2 Report
Comments and Suggestions for Authors
Davletbaeva and coworkers have reported a series of vapor-permeable polyurethanes based on ortho-phosphoric acid, PEG and triethanolamine. The formation of precursor AEPA-n-PEGs has been discussed which includes the etherification of OPA, particle size distributions and NMR spectra of products. The properties of polyurethanes derived from these AEPA-n-PEGs have been studied by methods such as stress-strain measurements, surface tension measurements, vapor permeability, TMA/DMA and AFM. While the overall results are promising and encouraging, several sections of the text lack the necessary information and data for publication and need to be addressed:
1) Page 4 Lines 126-127. What technique was used to calculate the OPA conversion at different time points (NMR etc.)? Details of the method must be included in the text.
2) Page 5 Lines 166-169. Where is the scheme showing the interaction in the TEOA-OPA-PEG reaction ? Figure 3 shows the kinetic plots. Should it be "Figure 4"? Please clarify.
3) Page 5 Lines 172-175. Is there any evidence in the literature for the formation of a crown-encapsulated proton ? Please provided references if so.
4) Page 5 Lines 176-179. What is the chemical evidence that all AEPA-n-PEGs have such a chemical structure? The "exact" structure has to be characterized thoroughly to determine its molecular mass. Has GPC/SEC been conduced on them ? Alternatively mass spectrometry measurements might also provide an exact mass for the AEPA-n-PEG species.
5) Page 6 Lines 183-184. The statement that particle sizes for AEPA-4-PEG are maximum is incorrect, as AEPA-3-PEG has the largest particle size distribution in acetone (Figure 5b (2)). Please correct this statement. Also, how were the particle sizes measured ? Light scattering ? If so, please add this to the text.
6) Page 6 Line 202. What is the splitting due to ? J coupling ? 2-D NMR experiments (COSY, HSQC, HMBC etc.) have to be conducted to elucidate the proton coupling and environment.
7) Page 7 Figure 6. What solvent was used to acquire the NMR spectra ? Please include this in the caption of the figure.
8) Page 7 Lines 216-217. In the 31P spectra, there are no peaks seen at "-1.4 and -1.5 ppm". Do the authors mean peaks at -0.4 and -0.5 ppm ? Please correct this in the text.
9) Page 7 Lines 222-224. What is the cause of the splitting of the 31P signal at 0.7 ppm ? A 2D 1H-31P correlation experiment must be conducted to clarify this interaction.
10) Page 7 Lines 225-232. Where are the results for AEPA-4-PEG and AEPA-5-PEG ? Please include them in Figure 7.
11) Where is the section on the formation of polyurethanes ? At least a few details must be presented such as the type of isocyanate used, and a general reaction scheme in a figure.
12) Page 8 Figure 8. How do the stress-strain curves of AEPA-n-PEG-PUs compare with commercial polyurethanes/ionomers ? Please include some comparisons and references in the text.
13) Page 8 Section 3.2. Have stress-strain curves been obtained for the polyurethanes obtained from AEPA-5-PEG at different reaction temperatures ? Please include them in the text as well.
14) Page 9 Line 259. Should it be "decrease in surface tension values for AEPA-5-PEG..." ? Please clarify.
15) Has GPC/SEC been conducted on the AEPA-n-PEG-PUs ? It is crucial to determine their molecular weight distributions for suitable applications.
16) In general for the figures, please use actual labels of the materials instead of numbers, as it is very confusing to keep track of different materials studied. A legend for colors could be used too.
Author Response
Thank you for your comments and feedback, which have helped us to substantially improve our manuscript. We have carefully studied your recommendations and prepared comments on each item you mentioned in the review.
1) Page 4 Lines 126-127. What technique was used to calculate the OPA conversion at different time points (NMR etc.)? Details of the method must be included in the text.
Answer: NMR spectroscopy has been added to the experimental part
2) Page 5 Lines 166-169. Where is the scheme showing the interaction in the TEOA-OPA-PEG reaction ? Figure 3 shows the kinetic plots. Should it be "Figure 4"? Please clarify.
Answer: Corrected.
3) Page 5 Lines 172-175. Is there any evidence in the literature for the formation of a crown-encapsulated proton? Please provided references if so.
Answer: Literary source has been added as reference [60]. Line 207.
4) Page 5 Lines 176-179. What is the chemical evidence that all AEPA-n-PEGs have such a chemical structure? The "exact" structure has to be characterized thoroughly to determine its molecular mass. Has GPC/SEC been conduced on them ? Alternatively mass spectrometry measurements might also provide an exact mass for the AEPA-n-PEG species.
Answer: Corrected. Line 208-210.
5) Page 6 Lines 183-184. The statement that particle sizes for AEPA-4-PEG are maximum is incorrect, as AEPA-3-PEG has the largest particle size distribution in acetone (Figure 5b (2)). Please correct this statement. Also, how were the particle sizes measured ? Light scattering ? If so, please add this to the text.
Answer: In connection with the comment, additions were made (line 213): «The particle sizes measured in water (Fig. 5a) turned out to be maximum for AEPA-4-PEG». Corrected Line 227-230. The experimental part added the dynamic light scattering method used to determine the particle size.
6) Page 6 Line 202. What is the splitting due to ? J coupling ? 2-D NMR experiments (COSY, HSQC, HMBC etc.) have to be conducted to elucidate the proton coupling and environment.
Answer: A detailed description of the NMR experiments добавлен in the experimental part.
7) Page 7 Figure 6. What solvent was used to acquire the NMR spectra? Please include this in the caption of the figure.
Answer: Added to the caption of the Figure 6: Solvent deuterated chloroform.
8) Page 7 Lines 216-217. In the 31P spectra, there are no peaks seen at "-1.4 and -1.5 ppm". Do the authors mean peaks at -0.4 and -0.5 ppm? Please correct this in the text.
Answer: Corrected
9) Page 7 Lines 222-224. What is the cause of the splitting of the 31P signal at 0.7 ppm ? A 2D 1H-31P correlation experiment must be conducted to clarify this interaction.
Answer: Unfortunately, we are currently unable to conduct such method.
10) Page 7 Lines 225-232. Where are the results for AEPA-4-PEG and AEPA-5-PEG ? Please include them in Figure 7.
Answer: According to the comment, advanced analysis of TGA curves has been added. Line 255-271.
11) Where is the section on the formation of polyurethanes ? At least a few details must be presented such as the type of isocyanate used, and a general reaction scheme in a figure.
Answer: According to the comment, the following description has been added to the experimental part « For the synthesis of polyurethanes, an aromatic polyisocyanate “Wannate PM-200” (PIC) purchased from Kumho Mitsui Chemicals, Inc. was used. (China)». Line 107-108.
On the line 274-277 the following description has been added: «For the synthesis of polyurethanes, AEPA-5-PEG and aromatic polyisocyanate (PIC) “Wannate PM-200” were used. The reaction of urethane formation is widely known [1-13], and in this work it involves the interaction of the terminal hydroxyl groups of AEPA-5-PEG and isocyanate groups of PIC.».
12) Page 8 Figure 8. How do the stress-strain curves of AEPA-n-PEG-PUs compare with commercial polyurethanes/ionomers ? Please include some comparisons and references in the text.
Answer: According to the comment, the following description has been added to the content: «It should be noted that the strength of AEPA-(3-5)-PEG is high and reaches values of 47-53 MPa. For comparison, the strength of polyurethane ionomers studied in [61] is in the range of 20-30 MPa.»… Line 284-286.
13) Page 8 Section 3.2. Have stress-strain curves been obtained for the polyurethanes obtained from AEPA-5-PEG at different reaction temperatures? Please include them in the text as well.
Answer: Stress-strain curves for the polyurethanes obtained from AEPA-5-PEG at different reaction temperatures were not obtained since it seemed more important to study the thermomechanical behavior of polyurethanes obtained from AEPA-5-PEG at different reaction temperatures (Fig. 12) and studying the patterns of changes in their vapor permeability.
14) Page 9 Line 259. Should it be "decrease in surface tension values for AEPA-5-PEG..." ? Please clarify.
Answer: An inaccuracy has been detected. In this regard, the content has been adjusted: «Measurements of surface tension isotherms (Fig. 10) made it possible to establish that with an increase in the synthesis temperature from 80 to 110 °C, the surface tension of AEPA-5-PEG solutions in water decreases. The obtained result confirms the conclusion that an increase in OPC conversion entails growing in the content of phosphate ions. ». Line 302-305.
15) Has GPC/SEC been conducted on the AEPA-n-PEG-PUs ? It is crucial to determine their molecular weight distributions for suitable applications.
Answer: GPC/SEC measurements on the AEPA-n-PEG-PUs were not conducted since the polymers are cross-linked and therefore insoluble. However, we measured the particle sizes for AEPA-n-PEG, presented in Figure 5.
16) In general for the figures, please use actual labels of the materials instead of numbers, as it is very confusing to keep track of different materials studied. A legend for colors could be used too.
Answer: Corrected.
Round 2
Reviewer 2 Report
Comments and Suggestions for Authors
The authors have made significant additions to certain sections and have made all the necessary changes to the manuscript, significantly improving its readability and quality. The work is suitable for publication in Polymers and should be of great interest to the polymer community in general.